# The isomer distribution of C₆H₆ products from the propargyl radical gas-phase recombination investigated by threshold-photoelectron spectroscopy

Helgi Rafn Hrodmarsson [1,2] ✉, Gustavo A. Garcia [1], Lyna Bourehil[1], Laurent Nahon [1], Bérenger Gans [3], Séverine Boyé-Péronne [3], Jean-Claude Guillemin [4] & Jean-Christophe Loison [5] ✉

The resonance-stabilization of the propargyl radical (C₃H₃) makes it among the most important reactive intermediates in extreme environments and grants it a long enough lifetime to recombine in both terrestrial combustion media and cold molecular clouds in space. This makes the propargyl self-reaction a pivotal step in the formation of benzene, the first aromatic ring, to eventually lead to polycyclic aromatic hydrocarbons in a variety of environments. In this work, by producing propargyl radicals in a flow tube where propyne reacted with F atoms and probing the reaction products by mass-selected threshold-photoelectron spectroscopy (TPES), we identified eight C₆H₆ products in total, including benzene. On top of providing the first comprehensive measurements of the branching ratios of the eight identified C₆H₆ isomers in the propargyl self reaction products (4 mbar, 298 K conditions), this study also highlights the advantages and disadvantages of using isomer-selective TPES to identify and quantify reaction products.

The presence of polycyclic aromatic hydrocarbons (PAHs) in nature represents a peculiar dichotomy of chemical relevance. Terrestrially PAHs play a key role as intermediates in the nascence and subsequent agglomeration of soot particles, the principal ingredient of particulate matter (PM)[1,2]. PM refers to dust particles created by incomplete combustion of gasoline, diesel, and coal, as well as those released by road construction and agricultural processes. An estimated 3.3 million deaths per year worldwide are attributed to atmospheric PM[3] and its negative influences on asthma, chronic obstructive pulmonary disease (COPD), lung cancer and global warming are all well documented[4]. PAHs themselves pose a risk to human health as many have been revealed to be toxic, mutagenic and/or carcinogenic[5].

In space, the presence of PAHs has been postulated since the 80's from the observation of mid-IR aromatic absorption bands[6,7], which are ubiquitous in various interstellar regions. In recent years, however, aromatic molecules such as ortho-benzyne[8], cyanobenzene[9], ethynylbenzene[10], cyanonaphthalene[11], indene[12,13] and others, have been definitively detected

in cold molecular clouds, on top of the benzene molecule which has been detected in circumstellar media[14] as well as Titan's atmosphere[15].

The process of bottom-up formation of PAHs at different temperature regimes is thus of interest to multiple fields, as PAHs can form in environments ranging from the very low temperatures of cold molecular clouds to the burning hot environments in combustion engines. Even from an environmental standpoint, understanding the fundamental reactions that lead to the formation of PAHs, and eventually, soot particles, is of paramount interest in order to minimize and even hinder their influence on terrestrial lifeforms and improve the quality of life on Earth.

The growth of PAHs is initiated by the formation of the first closed-ring species and is the quintessential step in incipient particle formation in aliphatic fuels[16]. A formation path toward benzene that garnered a lot of attention after its inception was the hydrogen-abstraction/acetylene-addition (HACA) mechanism, which was thought to be the dominant pathway toward PAH formation for a number of years[17]. More recently, however, data from combustion studies have revealed that the HACA mechanism

[1]Synchrotron SOLEIL, L'Orme des Merisiers, St. Aubin, F-91192 Gif sur Yvette, France. [2]Univ Paris Est Créteil and Université Paris Cité, CNRS, LISA UMR 7583, 94010 Créteil, France. [3]Université Paris-Saclay, CNRS, Institut des Sciences Moléculaires d'Orsay, F-91405 Orsay, France. [4]Univ Rennes, Ecole Nationale Supérieure de Chimie de Rennes, CNRS, ISCR – UMR6226, F-35000 Rennes, France. [5]Institut des Sciences Moléculaires, CNRS, Université de Bordeaux, F-33400 Talence, France. ✉e-mail: hhrodmarsson@lisa.ipsl.fr; jean-christophe.loison@cnrs.fr

only accounts for up to 6% of the phenanthrene and anthracene formation at temperatures between 1000 and 2000 K[18]. Other radical-radical reactions leading to benzene have been identified[19–23] but among the most efficient is the recombination of two propargyl ($C_3H_3$) radicals. A comprehensive historical perspective of research into the propargyl recombination reaction that covers the period up to 2005 can be found elsewhere[2,24]. Here we will highlight more recent developments in the last two decades or so.

The significance of the propargyl radical in terms of reactivity stems from the fact that its unpaired electron is delocalized amongst the three carbon atoms and hence propargyl has two stable resonance structures ($H_2C^•—C{\equiv}CH$ and $H_2C = C = {^•}CH$). This grants it a sufficiently long lifetime to survive in severe, high-temperature combustion environments, and in addition, significant reactivity to promote the recombination reaction. This property is further illustrated by the propargyl radical's reluctance to form stable bonds with molecules such as $O_2$ under high temperatures[24]. As a result, the increased concentration of propargyl radicals in flames allows the recombination reaction to become a relatively prominent reaction channel.

The propargyl radical itself has been considerably studied in terms of measuring its ionization energy and its absolute photoionization cross section[25–27]. It has been detected in the Taurus Molecular Cloud (TMC-1)[28] as well as its cation[29], and the presence of its cyclic, aromatic cation, c-$C_3H_3{^+}$, has been tentatively detected in the coma of Halley's comet[30].

Although the propargyl self-reaction is the simplest and most important of recombination reactions of resonantly stabilized free radicals[24], it is still very complicated. Miller and Klippenstein[31,32] managed to fit the most pertinent features of the reaction into several panels of potential energy surfaces (see Fig. 2 in Miller and Klippenstein[32]) that were later summarized in a kinetic model by Tang et al.[33,34].

In brief, the propargyl recombination reaction can be divided into three phases that are displayed in Fig. 1. For the purposes of comparison, Fig. 1 correlates the naming convention of the different isomer products introduced in this work, with that of Miller and Klippenstein[31,32]. During phase 1, two propargyl radicals can form three different isomers, 1,2-hexadiene-5-yne (12HD5Y), 1,2,4,5-hexatetraene (1245HT), and 1,5-hexadiyne (15HDY). 15HDY isomerizes back into 1245HT within phase 1. The 3,4-dimethylenecyclobutene (34DMCB) isomer can be considered as a surrogate for the 'tail-to-tail' entrance channel that forms 1245HT. Indeed, it has been shown that, while 1245HT is efficiently and predominantly converted to 34DMCB[33], it also isomerizes to 1,3-hexadiene-5-yne (13HD5Y) which appears in the next phase. During phase 2, 12HD5Y can isomerize to 2-ethynyl-1,3-butadiene (2E13BD), while 34DMCB can isomerize to form either fulvene or 13HD5Y, the latter through 1245HT. In phase 2, 2E13BD and fulvene equilibrate. In phase 3, benzene is formed via isomerization of 13HD5Y or through the equilibrium between fulvene and benzene.

Experimental validation of the above-mentioned model and quantification of the benzene exit channel is challenging due to the complexity of the propargyl self-reaction and the number of possible intermediates and final products. A universal, method of detection is necessary to account for multiple $C_6H_6$ species simultaneously, in situ and in real-time. Mass spectrometry coupled to tunable vacuum ultraviolet (VUV) radiation from synchrotron sources (known as photoionization mass spectrometry) has been used successfully in the past for species determination in complex media[35–37]. Recently, Zhao et al.[38] used photoionization efficiency curves to tentatively infer the formation of benzene molecules from the propargyl self-reaction at high temperature along with a more clear detection of three of the other structural isomers, namely fulvene, 15HDY, and 2E13BD. A shortcoming of using photoionization efficiency curves for isomer separation is that these curves, particularly for the species in question, look exceedingly similar which makes disentangling a total ion yield comprising multiple isomers problematic when the isomers have ionization energies close to each other.

A more sensitive method of isomer detection in mass spectrometric studies is using mass-selected photoelectron spectroscopy (PES), either at fixed photon energy[39] or with the threshold PES method (TPES)[40,41]. The former has the multiplex advantage since the photon energy is not scanned, which is typically faster and also advantageous if experimental conditions are not stable. The latter offers a better, constant, energy resolution but requires scanning the photon energy and therefore is more time-consuming. Recently, Savee et al.[42] used this technique to study the uni-molecular isomerization of 1,5-hexadiyne (15HDY) which is one of the three adducts of the prompt propargyl self-reaction. They were able to measure the branching ratio of fulvene, benzene and 34DMCB production. This study confirmed and refined Stein's previous results on the study of low-temperature isomerization of 15HDY[39]. It also corroborated the global analysis of Miller and Klippenstein results[32], testing, however, only a part of the global potential energy surface of the $C_3H_3 + C_3H_3$ reaction and requiring adjustment of the energy of some transition states to reproduce the results.

In this work, we have employed a microwave discharge flow-tube reactor to produce $C_3H_3$ radicals in a continuous and stable manner. The flow-tube was coupled to a synchrotron VUV radiation source at the French national facility SOLEIL, and the mass-selected TPES for $C_6H_6$ reaction products was recorded via double imaging photoelectron/photoion coincidence ($i^2$PEPICO) techniques. To identify and disentangle all isomeric products, individual PES or TPES of each isomer were acquired or simulated separately. Besides the primary product of interest, benzene, we found that we also detected most of the reaction intermediates outlined in Fig. 1, namely, 12HD5Y and 1245HT from phase 1, 34DMCB, 2E13BD, fulvene, and 13HD5Y from phase 2.

## Results and discussion
### Composition of the reactor

The mass spectrum corresponding to the composition of the reactor was obtained from the ion time-of-flight (TOF) spectrum integrated over the photon energy from 8.20 to 10.12 eV for the F + propyne scheme (Fig. 2). This spectral range was used to study the photoionizing transitions of the $C_6H_6$ isomers (m/z 78) from their neutral ground state to their cationic ground state.

The two most intense peaks in Fig. 2 are assigned to $C_3H_3{^+}$ at m/z 39 and $C_3H_4F^+$ at m/z 59 which correspond to the following exothermic reactions for the neutral species:

$$CH_3CCH + F \rightarrow C_3H_3 + HF \tag{1}$$

$$CH_3CCH + F \rightarrow C_3H_4F \tag{2}$$

For the experimental conditions where atomic F density is low and the reaction time is equal to 2 ms, several weaker peaks appear corresponding to addition reactions between radicals:

$$C_3H_3 + C_3H_3 \rightarrow C_6H_6 \; m/z \; 78 \tag{3}$$

$$C_3H_3 + C_3H_4F \rightarrow C_6H_7F \; m/z \; 98 \tag{4}$$

$$C_3H_4F + C_3H_4F \rightarrow C_6H_8F_2 \; m/z \; 118 \tag{5}$$

Note that our mass resolution does not allow to distinguish the members of isobaric families. Contamination by $C_3H_4F_2$, corresponding to the addition of two fluorine atoms on the parent species, must also be considered to account for the signal at m/z 78. In the Supplementary Information (Supplementary Note 1) we show that, for low atomic F concentration, the contribution of fluorine-bearing structures to the signal observed for m/z 78 is negligible.

### $C_6H_6$ isomers

Figure 3 shows the TPES for m/z 78 obtained from the $C_3H_3 + C_3H_3$ reaction at 4 mbar, in the range 8.20-10.12 eV. To extract mole fractions of the different $C_6H_6$ isomers contributing to the TPES, we need to know the

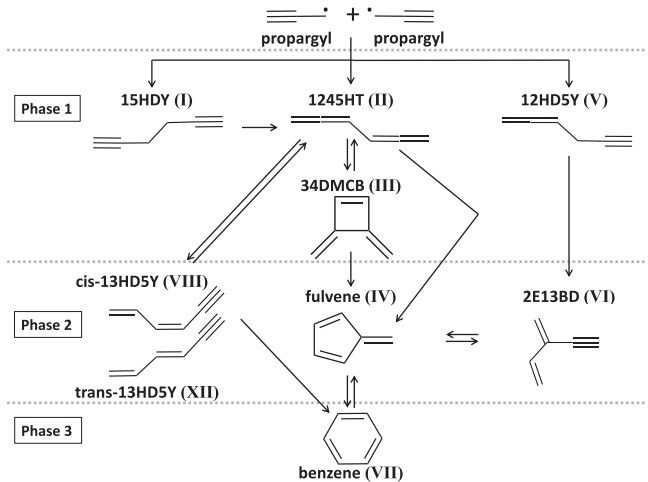

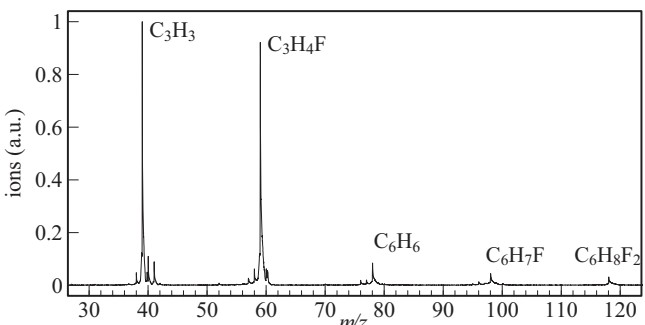

**Fig. 1 | $C_3H_3 + C_3H_3$ reaction scheme leading eventually to benzene.** Illustration of the reaction routes toward benzene through the propargyl recombination reaction. The naming convention used by Miller and Klippenstein[31,32] for the different $C_6H_6$ isomers is recalled in parenthesis with Roman numerals.

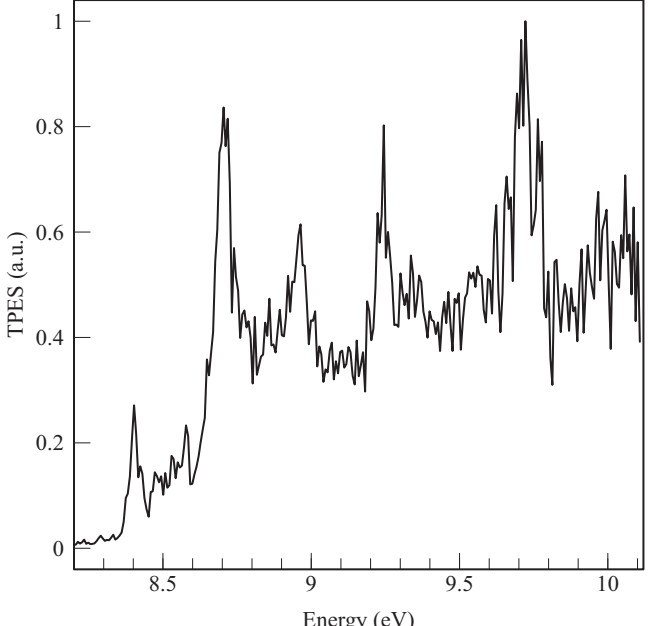

**Fig. 2 | TOF mass spectra of flow tube products.** TOF mass spectra integrated between 8.20 and 10.12 eV photon energy for the $F + CH_3CCH$ system with $[F] \ll [CH_3CCH]$.

**Fig. 3 | TPES for $m/z$ 78.** Experimental threshold-photoelectron spectrum recorded for $m/z$ 78 with a total resolution of 20 meV.

TPES of individual various isomers (so-called reference TPES, presented below), as well as their photoionization cross sections.

## Reference TPES

We consider all isomers calculated by Miller and Klippenstein[32] to be potentially produced in the flow-tube reactor, excluding unstable species (species numbered **IX, X,** and **XI** in Miller and Klippenstein[32] which correspond to $o$-, $m$-, and $p$-dihydrophenyl radicals). A number of experimental PES have already been recorded with variable resolutions for 15HDY (**I**)[42–44], 1245HT (**II**)[44], 34DMCB (**III**)[42,45], fulvene (**IV**)[42,44], 12HD5Y (**V**)[44], benzene (**VI**)[39], and indirectly a mixture of cis- and trans-13HD5Y (**VIII** and **XII**)[42]. For the missing (or ambiguous) PES, the isomers 2E13BD (**VI**), and cis- and trans-13HD5Y (**VIII** and **XII**) have been synthesized prior to the experimental campaign, and we have measured their TPES or PES at fixed energy. The resulting experimental spectra are shown in Fig. 4, together with the simulated ones in color.

**1,5-hexadiyne.** H-C≡C-CH$_2$-CH$_2$-C≡CH, 15HDY, species **I** in Miller and Klippenstein[32]:

The PES have been recorded by Brogli et al. [43], and Bischof et al. [44], and the TPES has been published recently by Savee et al. [42]. This species cannot be fully observed in our experiment because its first ionization onset (9.90 eV) is just before the upper limit of our energy range and the maximum of its TPES (10.48 eV) falls outside of our energy range. Hence, we do not consider this isomer in our simulated spectrum. However, note that the simulation of the PES of the 15HDY is complicated as it involves several states (the ground state of the cation, as well as the two lowest excited states are very close in energy). The formation of 15HDY is expected from a 'head-to-head' recombination of propargyl radicals and was predicted to generally be the product with the highest yield at high temperatures and pressures[32], but only a minor species at room temperature and low pressure corresponding to those used in our experiment. It was detected by Zhao et al.[38] with a small yield.

**1,2,4,5-hexatetraene.** H$_2$C = C = CH-CH = C = CH$_2$, 1245HT, species **II** in Miller and Klippenstein[32]:

The PES of 1245HT was previously recorded by Bischof et al. [44]. This isomer was present during the synthesis of 12HD5Y (see section 'Methods') so we were able to measure its TPES simultaneously (see panel a) of Fig. 4). We obtained a good agreement with the results of Bischof et al., our ionization energy (IE) being measured equal to 8.494 ± 0.010 eV (compared with 8.53 eV from Bischof et al.[44]). For 1245HT, the Franck-Condon simulation (green curve in Fig. 4a)) is in good agreement with the experimental spectrum, thus we use the calculated spectra to reproduce the contribution of 1245HT in our experimental TPES of the $C_3H_3 + C_3H_3$ reaction.

**3,4-dimethylene-cyclobutene.** H$_2$C = (c-C$_4$H$_2$) = CH$_2$, 34DMCB, species **III** in Miller and Klippenstein[32]:

The PES of 34DMCB was recorded by Heilbronner et al.[45] who measured the IE as 8.80 eV. Savee et al.[42] measured its TPES recently. In Fig. 4 panel b), we report the spectrum of Savee et al. convolved with a Gaussian function to get a linewidth of 20 meV, corresponding to our experimental resolution. It should be noted that the 34DMCB TPES cannot be reproduced by a standard Franck-Condon model[42] and hence we cannot use this method to reproduce its contribution in the spectrum of Fig. 3. We then use the spectrum of Savee et al. convolved with a Gaussian function to get a linewidth of 20 meV to reproduce its contribution in our experimental TPES of the $C_3H_3 + C_3H_3$ reaction.

**Fulvene.** c-C$_5$H$_4$ = CH$_2$, species **IV** in Miller and Klippenstein[32]:

The PES of fulvene has been recorded by Bischof et al. [44], and its TPES has been recently published by Savee et al. [42]. In this latter work, the IE of fulvene was determined as 8.398 eV, which is well reproduced in our calculations. The fulvene TPES in the region 9.5–10.1 eV contains transitions to

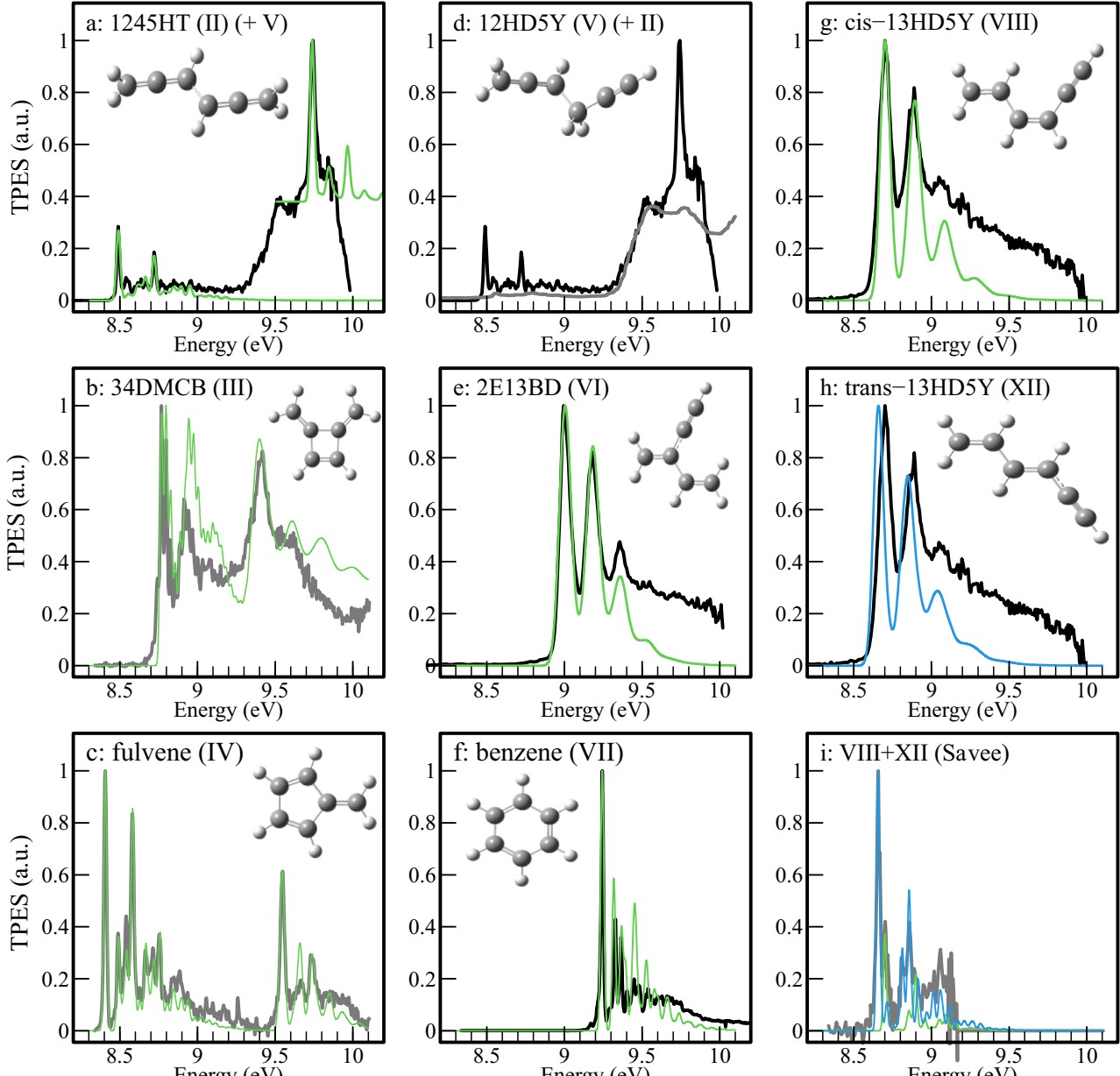

**Fig. 4 | Comparisons of TPES and PES of C₆H₆ isomers.** Compilation of the TPES and PES of the various $C_6H_6$ isomers used to identify these products in the propargyl recombination reaction. For the experimental spectra, panels **a**, **d**, **e**, **g** and **h** correspond to spectra measured in the present work (shown in black). Panels **b**, **c** and **i** (dark grey curve) are TPES taken from Savee et al.[42] Panel **f** is the PES from Heilbronner et al.[45] In panel **d**), the experimental PES of Bischof et al.[44] is displayed in dark grey for identification purposes. Calculated spectra are displayed in green or in blue (panels **h**) and **i**) to separate isomers **VIII** and **XII**). See section "Reference TPES" for details on reference spectra.

the first electronically excited state. Here, we use the experimental spectrum of Savee et al. convolved with a Gaussian function with a FWHM of 20 meV to reproduce its contribution in our experimental TPES of the $C_3H_3 + C_3H_3$ reaction. The resulting spectrum is displayed in panel c) of Fig. 4.

**1,2-hexadiene-5-yne.** $H_2C = C = CH\text{-}CH_2\text{-}C \equiv CH$, 12HD5Y, species **V** in Miller and Klippenstein[32]:

The PES of 12HD5Y was recorded by Bischof et al.[44] and is mostly featureless. Hence, we recorded a TPES of a synthesized sample of 12HD5Y (see panel d) of Fig. 4, black curve) which also contained impurities of 1245HT, a co-product of the synthesis. The spectrum is in good agreement with that of Bischof et al.[44], (see Fig. 4d), dark grey curve) the lack of structure being merely a manifestation of an important change in geometry between the neutral and the ion. As there are no resolved bands, the

determination of the effective ionization cross-section is more difficult because the theoretical spectrum of this species cannot be simulated with Gaussian. We use the spectrum of Bischof et al.[44] to reproduce its contribution in our experimental TPES of the $C_3H_3 + C_3H_3$ reaction.

**2-Ethynyl-1,3-butadiene.** $HC \equiv C\text{-}C( = CH_2)\text{-}CH = CH_2$, 2E13BD, species in **VI** in Miller and Klippenstein[32]:

For 2E13BD, there are two conformers that can be formed. No prior PES was known so it was synthesized, isolated, and its TPES measured in the present work (see panel e) of Fig. 4). The TPES gave an ionization energy of 9.007 ± 0.007 eV, which agrees very well with the calculations. The second conformer, which is calculated to appear 12 kJ/mol higher in energy, was not observed in the synthesized sample, and its ionization energy is computed at 10.37 eV. Therefore, it is not considered further in this work.

**Benzene**. species **VII** in Miller and Klippenstein[32]:

The PES of benzene is well-known with a recorded IE of 9.24 eV[39]. Our calculated spectrum convincingly replicates that of Heilbronner et al.[45] using a FWHM of 15 meV (see panel f) of Fig. 4). The reference spectrum used here is the experimental spectrum from Heilbronner et al.[45] convolved with a Gaussian function to get a linewidth of 20 meV, corresponding to our experimental resolution.

**Cis- (Z) & trans- (E) 1,3-hexadiene-5-yne.** $CH_2 = CHCH = CHC \equiv CH$, cis- & trans-13HD5Y, species **VIII** and **XII** in Miller and Klippenstein[32]:

Two isomers of 13HD5Y exist where the terminal vinyl and ethynyl groups are either found in a cis- or a trans-arrangement on either side of a double bond. Hence, isomer **VIII** corresponds to the cis-conformer (cis-13HD5Y) and isomer **XII** the trans-conformer (trans-13HD5Y). A transition state exists between the two isomers, which was calculated as 205 kJ/mol above the minimum of the cis-conformer[32]. The TPES of a mixture of both isomers has been recently recorded by Savee et al.[42] (see dark grey curve in panel i) of Fig. 4) Here, time constraints related to synchrotron access prevented us from recording a TPES, and the PES resulting from both conformers was recorded instead at a photon energy of 10.1 eV (see panels g) and h) of Fig. 4). Unfortunately, the lower resolution achieved for faster photoelectrons precludes the separation of the two isomers, in contrast to the work of Savee et al.[42]. Both isomers are considered for identification in our experiment but, as the proportion of each isomer is not controlled by temperature but by their chemical production, it is not possible to use the spectrum of the mixture obtained by Savee et al. Hence, we use individual Franck-Condon simulations which agree rather well with the experimental spectra. We reproduce the Savee et al.[42] spectra with a **XII/VIII** ratio equal to 3, and our TPES spectra with solely the **VIII** isomer, perhaps due to the adiabatic expansion cooling our sample down to a few tens of K. We then use the spectrum of Savee et al. convolved with a Gaussian function to get a linewidth of 20 meV to reproduce its contribution in our experimental TPES of the $C_3H_3 + C_3H_3$ reaction.

**1,5-hexadiene-3-yne.** $CH_2 = CHC \equiv CC = CH_2$, 15HD3Y, species **XIII** in Miller and Klippenstein[32]) & **1,2,3,5-hexatetraene** ($CH_2 = C = C = CHCH = CH_2$, 1235HT, species **XIV** in Miller and Klippenstein[32]:

Finally, we also consider isomers **XIII** and **XIV**, 15HD3Y and 1235HT, respectively, for completeness. These two isomers have not been accounted for in some previous works[32]. Through successive H-migration of two H atoms, 15HD3Y can be formed from 1245HT (isomer II). This process takes place through a transition state localized -50 kJ/mol below the $C_3H_3 + C_3H_3$ entrance channel. 1245HT can also be formed through a transition state localized -26 kJ/mol below that same channel. However, as the transition states producing these two isomers are much higher than other transition states leading to the other isomers presented, we do not include these species in our analysis like in the work of Miller and Klippenstein[32].

## Identification of $C_6H_6$ isomer products in the propargyl recombination reaction

Figure 5 shows in black the measured TPES between 8.20 and 10.12 eV corresponding to the m/z 78 signal recorded at optimum flow tube conditions to produce propargyl ($C_3H_3$) radicals. As mentioned earlier, the reference spectra of individual $C_6H_6$ isomers are built from Franck-Condon simulations matching the experimental spectra, except for 12HD5Y which cannot be easily calculated. They are presented in green in Fig. 5 and compared to the total signal. In Fig. 5 panel i), the individual references have been added according to their optimized molar fractions and are compared to the m/z 78 TPES. The optimization is performed through a least-squares fit of the reference spectra to the experimental TPES.

The agreement between the optimized sum of the references and experimental spectra is, for the most part, excellent, which leads to the unambiguous identification of benzene, fulvene, 1245HT, 12HD5Y,

34DMCB and 2E13BD (panels f), c), a), d), b) and e) of Fig. 5). The separation of cis- and trans-13HD5Y (see panels g) et h) of Fig. 5) is more challenging since their TPES are very similar but are clearly shifted by approximately 20 meV. Using only one of them does not replicate the strong resonance at 8.7 eV convincingly, while utilizing both does a very good job at replicating its shape and its intensity.

Despite the overall agreement, there are several notable issues. Firstly, the peak at 8.9 eV in the experimental TPES is not perfectly reproduced. Secondly, the peak at 9.7 eV is only reproduced by 1245HT, which has a small peak at 8.5 eV that is only partially visible in our spectrum. It should also be noted that the large number of isomers produced makes interpretation of the spectrum complicated, particularly for isomers with no easily distinguishable lines.

Additionally, differences in wavelength-dependent photoionization cross sections, notably due to continuum resonances such as autoionizations, could account for the lack of agreement in some of the TPES features.

## Branching ratio determination for the dimerization of $C_3H_3$ and discussion

To estimate the branching ratios of the $C_3H_3 + C_3H_3$ reaction leading to the different $C_6H_6$ isomers and their error bars, the mixture spectrum (black spectrum depicted in the panels of Fig. 5) must be reproduced summing all the reference spectra (green spectra of Fig. 5) with appropriate weighting factors. To achieve this, we used a Monte-Carlo approach where we first adjusted initial weights manually to reproduce the mixture spectrum as well as possible (this is depicted with a red curve in Fig. 6). Then, for each reference spectrum, we randomly draw a complete set of all weight values within uniform distributions spanning their initial values ± 0.2. Then, with the new set of weights the spectra are summed, and the obtained spectrum is compared with our mixture spectrum. To choose whether the new summed spectrum is accepted or rejected, we apply an arbitrary criterion that rejects the newly summed spectrum if the average of the absolute difference between the experimental and new summed spectrum is smaller than 9.2% up to 9.9 eV. We choose this criterion because the experimental spectrum is not well reproduced with the reference spectra at high energy where a numerical criterion value in the 9–10% range does not critically change the spectrum. There is a continuous increase above 9.9 eV in the mixture spectrum that is not successfully replicated in the summed spectrum potentially due to the fact that we did not include the 15HDY in our analysis. The simulated weighting factors and their statistical distributions are shown in the Supplementary Information (Supplementary Note 2) as well as the comparison of the calculated spectra using the branching ratios taken from Miller and Klippenstein[32].

To estimate the molar fractions of the various isomers, and given the lack of information on their absolute photoionization cross-sections, we follow the same procedure as for scenario 1 of Savee et al.[42] and consider that the integral of the TPES is equal to the number of electrons involved in the transition, i.e., 2 for each transition except for benzene because the ground state of the cation is doubly degenerate (involving then 4 electrons). To calculate the exact number of transitions involved in the PES we use the Electron Propagator Theory (EPT) with Gaussian16 (in most cases the transitions are well separated except for 12HD5Y). A potential source of error with this approach is the presence of autoionizations, which are known to affect the TPES and thus can skew the estimated molar fractions.

We have estimated the photoionization cross section of 12HD5Y considering that there were two transitions between 8.3 and 9.9 eV involving two electrons for each transition (ionization toward the ground state and toward the first excited state of the cation as determined by Electron Propagator Theory). This photoionization cross-section estimate is consistent with the 3/2 ratio between 12HD5Y and 1245HT in the mixture we used, a ratio determined by their NMR signature.

Given the aforementioned limitations and the lack of perfect agreement for some of the TPES features related to benzene and 1245HT, the former might be underestimated, and the latter overestimated. Considering our hypothesis on the effective photoionization cross sections and the

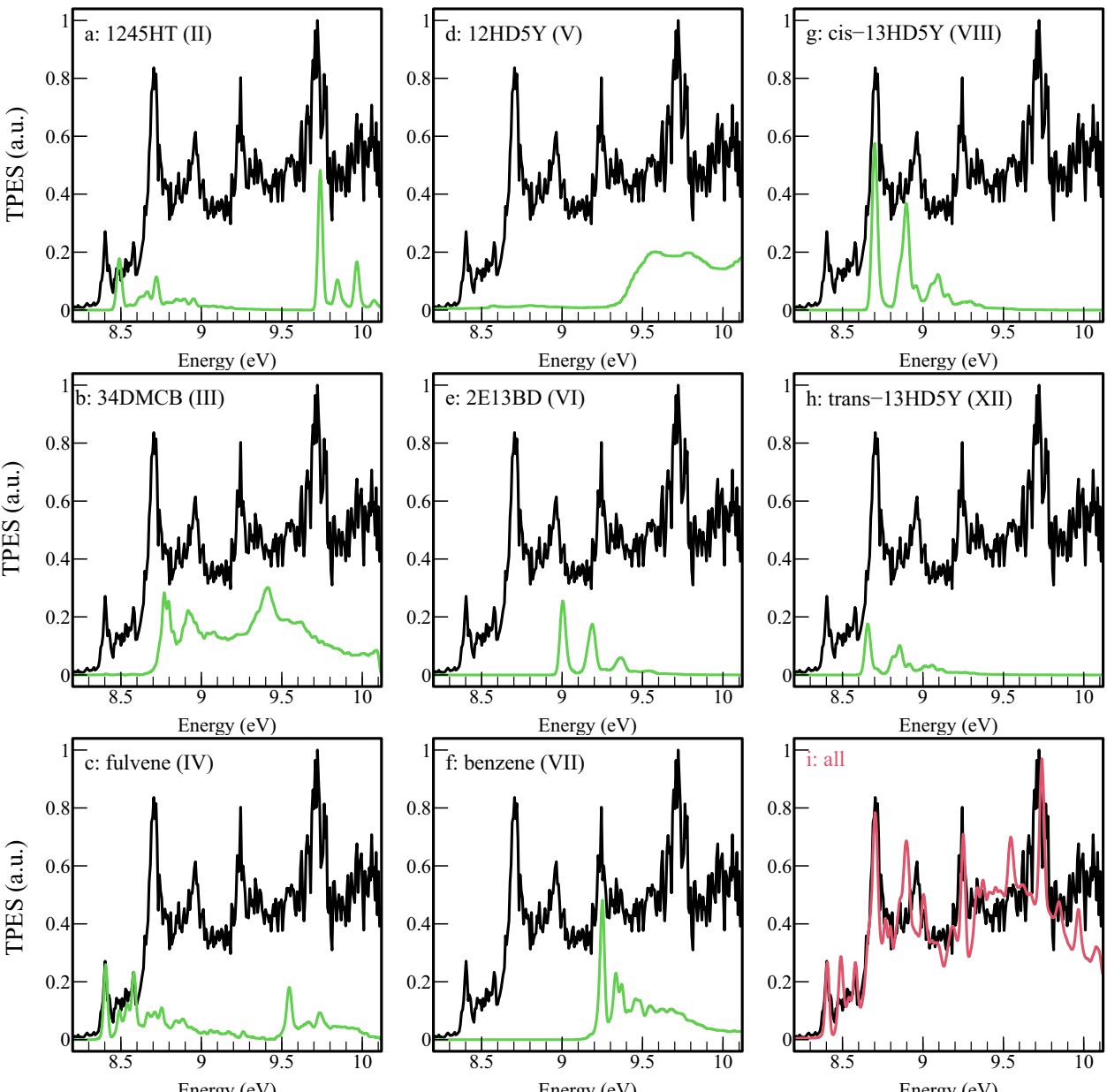

**Fig. 5 | Reference TPES used to replicate experiment.** Experimental TPES of *m/z* 78 in black, and comparison with the reference TPES of **a**) 1245HT (**II**), **b**) 34DMCB (**III**), **c**) fulvene (**IV**), **d**) 12HD5Y (**V**), **e**) 2E13BD (**VI**), **f**) benzene (**VII**), **g**) cis-13HD5Y (**VIII**), **h**) trans-13HD5Y (**XII**) outlined above (green curves). In panel **i**) shown in red is the weighted sum of all contributing reference spectra with the branching ratios listed in Table 1.

simulated weighting factors used to reproduce the *m/z* 78 spectrum, the branching ratios or the different isomer channels have been estimated at 4 mbar of total pressure, and they are reported in Table 1. Note that we have normalized to 100% on the species observed without taking into account 15HDY (**I**) or the phenyl + H dissociation channel which are calculated to be relatively minor products according to Miller and Klippenstein[32], accounting for around 18% of the total. Comparison with previous experimental results is difficult, given the variability of the experimental conditions (temperature and pressure), particularly for the recent experiment by Zhao et al.[38] carried out at 270 mbar. Therefore, only a comparison with the branching ratios derived from Miller and Klippenstein[32] calculations (4 mbar and room temperature) is proposed in Table 1. A comparison of our experimental TPES with the weighted sum of all contributing reference spectra with the branching ratios from the Miller and Klippenstein study[32] is presented in Fig. S7.

Overall, the agreement between our results and the predictions is qualitatively good, although there are some notable differences. We observe a wide variety of isomers, with benzene production being a minor channel for our conditions of low pressure and ambient temperature. There are, however, quite a few important differences between the branching ratios deduced from our spectrum and the prevision of Miller and Klippenstein[32] as well as previous results. One of the reasons is certainly our approximation of the value of the ionization cross sections. Another reason is the large uncertainties for isomers without a well-structured spectrum. However, there are a few issues that are left unexplained outside our experimental uncertainties.

One is the underestimation of fulvene production. This disagreement is surprising because the identification of fulvene and benzene in our TPES is unambiguous and allows comparison between our results, those of Savee et al.[42] and the predictions of Miller & Klippenstein[32]. Performing their

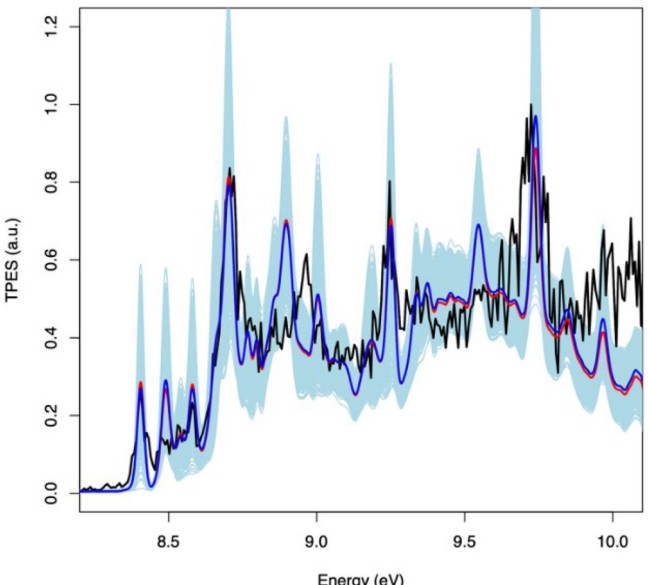

**Fig. 6 | Simulated TPES replicating experiment.** Experimental TPES of *m/z* 78 (black). Initial summed spectrum with manually adjusted weights of the individual reference spectra (red). Summed spectrum with simulated weights (blue). The light-blue shaded area corresponds to the superposition of the generated spectra from the Monte-Carlo procedure (see section on branching ratio determinations for details and Fig. S6 of the Supplementary Note 2 for histograms of the simulated weights).

**Table 1 | Branching ratios of the $C_3H_3 + C_3H_3$ reactions experimentally determined in this work at 4 mbar and room temperature and compared with the theoretical results of Miller and Klippenstein[32]**

| Species | Branching ratio | |
|---|---|---|
| | Exp. (this work) | Calc. (Ref. 34) |
| 1245HT (**II**) | 7 ± 4% | 8% |
| 34DMCB (**III**) | 23 ± 8% | 16% |
| Fulvene (**IV**) | 14 ± 5% | 26% |
| 12HD5Y (**V**) | 13 ± 6% | 2% |
| 2E13DB(**VI**) | 8 ± 5% | 28% |
| Benzene (**VII**) | 11 ± 5% | 6% |
| cis-13HD5Y (**VIII**) | 18 ± 6% | 7% |
| trans-13HD5Y (**XII**) | 6 ± 5% | 7% |

experiment at 748 K, Savee et al. obtained a signal ratio of fulvene:benzene of 2:1, which translates to an abundance ratio of 4:1 after correcting for the estimated photoionization cross sections. This ratio is in relatively good agreement with the predictions of Miller & Klippenstein for the temperature and pressure in the experiment, with the barrier connecting the wells 15HDY(**I**) and 1245HT(**II**) increased by 1.0 kcal/mol. In our spectrum, the signals for fulvene and benzene lead to a fulvene/benzene abundance ratio of 1.3, smaller than the values of Miller & Klippenstein, which were around 3.5 at the temperature and pressure used in our experiment. However, the agreement might improve if we apply the modifications to the transition state values made by Savee et al.[42].

The stronger disagreements between our experiment and the work of Miller & Klippenstein is for 12HD5Y (**V**) and 2E13DB (**VI**), both linked to the head-to-tail addition of two propargyl moieties. The production of these two isomers seems to be reversed in our experiment versus the Miller & Klippenstein prediction, with the sum of these two species, corresponding more or less to the branching ratios of the head-to-tail addition, in relatively good agreement with the prediction. It should be noted also that for 12HD5Y (**V**) the PES is not well structured, and the branching ratios are then highly uncertain. The large uncertainty values of the branching ratio stem from the non-structured signal in the 9.5–10.0 eV region. For the 2E13DB (**VI**), on the other hand, the TPES is well structured (see Fig.4 e))— determined in this work thanks to a separate synthesis. The absence of peaks at 9.0 and 9.18 eV in our spectrum clearly supports the low abundance of **VI**, in contrast to the results of Miller & Klippenstein[32] for which 2E13DB (**VI**) is one of the major products at low pressure. This implies one of two things. Either the production of 2E13DB (**VI**) is very low if the head-to-tail addition is disfavored compared to the head-to-head and tail-to-tail additions (considering our value for 12HD5Y (**V**) as an upper limit), or the transition state toward fulvene is lower than the value calculated by Miller & Klippenstein[32]. This also seems in partial contradiction to the results of Tang et al.[33] who measured the $C_6H_6$ product distributions at 25 and 50 bar over 720-1350 K and detected 12HD5Y (**V**) but also 2E13BD (**VI**) as one of the main products. They used propargyl iodide as a precursor in a high-pressure single-pulse shock tube and detected the various products through gas chromatography which is not an in situ method. They determined the entry

branching ratios through modeling the kinetics of the detected $C_6H_6$ isomers using a semi-detailed model updated one year later[34]. They obtained a branching ratio for the antisymmetric addition product 12HD5Y (head-to-tail addition) equal to 38%, and for the symmetric additions they found 44% for 15HDY (**I**) (head-to-head addition), and 18% for 1245HT (**II**) (tail-to-tail addition). If we consider the sum of the 12HD5Y (**V**) and 2E13BD (**VI**) isomers, our results are not necessarily in complete contradiction with Miller & Klippenstein[32] and Tang et al.[33,34], but given the potential energy surface, it seems very unlikely that 12HD5Y (**V**) could be stabilized at 4 mbar to explain our results. Indeed, 12HD5Y (**V**) seems to be stabilized at higher pressure as determined by Fahr and Nayak[46] who obtained a branching ratio for 12HD5Y (**V**) equal to 25% at room temperature and 66.7 mbar, a value in good agreement with the prediction of Miller & Klippenstein[32].

Another, minor, disagreement is the ratio of abundances between the two rotamers of 13HD5Y (**VIII** and **XII**) produced through isomerization of 1245HT (**II**), which is in turn generated either during tail-to-tail addition or after isomerization of 15HDY(**I**) after head-to-head addition. As seen in Fig. 1, the production of 13HD5Y (**VIII** and **XII**) competes with the production of 34DMCB (**III**), fulvene (**IV**) and benzene (**VII**), the fulvene being calculated to dominate at low pressure. The 13HD5Y (**VIII** and **XII**) were observed by Savee et al.[42] during the isomerization of 15HDY (**I**) at high temperature to simulate the addition pathway. The first stage in the evolution of 15HDY(**I**) is the isomerization to 1245HT(**II**) but was not seen at high temperature because 1245HT (**II**) isomerizes immediately to 34DMCB (**III**), which is the main pathway toward formation of fulvene and benzene, the 13HD5Y (**VIII** and **XII**) being a minor product. In our experimental conditions—starting with $C_3H_3$ and an unheated reactor—we observe 1245HT (**II**) and 13HD5Y (**VIII** and **XII**) in greater abundance with the sum of both isomers in good agreement with the predictions of Miller & Klippenstein[32]. However, the ratio between the two isomers that we obtained, with cis-13HD5Y (**XII**) being three times more abundant than trans-13HD5Y (**VIII**), disagrees with the theoretical predictions. This disagreement cannot be explained by the position of the transition state between 1245HT(**II**) and trans-13HD5Y (**VIII**) or the transition state between trans-13HD5Y (**VIII**) and cis-13HD5Y (**XII**), as otherwise trans-13HD5Y (**VIII**) should be more abundant or both 13HD5Y (**VIII** and **XII**) should be less abundant. Further, it cannot be due to our ionization cross-section approximation, as both should be very similar as a function of the very close electronic properties between the two isomers. The rotamers contribution to *m/z* 78 is mainly fixed by the peak at 8.7 eV (see right column in Fig. 5). Because of their similar ionization energies (relative shift of only 44 meV), we have simulated the rotational envelopes of each isomer for a more precise quantification. We have applied the calculated rotational broadening to a Gaussian function centered on the adiabatic ionization energies of each isomer deduced from the spectra of Savee et al. (8.692 eV for **VIII** and 8.648 eV for **XII**). It should be noted, however, that a shift of

20 meV toward the blue of our spectrum (or toward the red for the spectrum of Savee et al.) completely changes the proportions, with similar abundances for the two isomers in better agreement with Miller & Klippenstein. This shift seems, however, too large given the calibration methods and the close agreement, within few meVs, with well-known systems such as benzene or fulvene.

## Conclusions

We have measured the isomeric branching ratios of eight $C_6H_6$ products, including benzene, from the $C_3H_3$ self-reaction at 4 mbar and room temperature using the i²PEPICO technique. To identify all the $C_6H_6$ products formed in the $C_6H_6$ chemical network, we also synthesized and recorded the spectra of 2-ethynyl-1,3-butadiene and 1,2,4,5-hexatetraene which allowed us to determine their ionization energies, *viz.*, $9.007 \pm 0.007$ eV and $8.494 \pm 0.010$ eV, respectively. The data show that DFT calculations using the M06-2X functional followed by a Franck-Condon simulation lead to a remarkably good agreement with experimental photoelectron spectra both for linear and cyclic isomers, in line with the recently published benchmark on its ability to predict adiabatic ionization energies of PAHs[47]. The quality of the simulated spectra is sufficient to be used as fingerprints to identify products in complex gas phase reactions.

Our measured branching ratios for the $C_6H_6$ products are in relatively good agreement with the predictions of Miller & Klippenstein[32]. The quantitative discrepancies observed outside experimental uncertainties could be explained by small adjustments in the transition states of the different isomerization pathways involved as well as by our rough estimate of the photoionization cross-sections.

The use of the TPES technique for product detection has been gaining momentum for the past decade and we hope that current limitations, related to lack of high-quality TPES and absolute photoionization cross-sections, will be overcome in the near feature with the recent emergence of data bases[48].

## Methods
### Experimental

The experimental setup has been described in detail in a previous work[49] and only a brief description is given here. Pure propyne from Air Liquide was swept into a quartz reactor with a continuous flow of helium (1000 sccm, standard cubic centimeters per minute), while fluorine atoms were produced by microwave discharge of a 5% $F_2$/He mixture and fed into a quartz shower-head injector that slides into the center of the reactor. The final pressure in the reactor was 4 mbar. The concentrations of $C_3H_4$ and F were controlled *via* mass flow controllers and estimated at $1 \times 10^{15}$ and $1 \times 10^{13}$ molecules.cm⁻³, respectively. We used a high concentration of excess propyne compared to atomic fluorine to minimize side reactions with fluorine. The contents of the reactor were then expanded through a 1 mm Teflon skimmer into an intermediate vacuum chamber[50] ($10^{-4}$ mbar). The distance between injector and skimmer gave the reaction time which was of the order of a few milliseconds. The molecular beam then traversed a second skimmer before reaching the center of the DELICIOUS3 i²PEPICO spectrometer[51] where it intersected the synchrotron beam. For acquisition of the reference spectra of each isomer, the species were synthesized (see paragraph below), placed in a cooled bubbler and adiabatically expanded through a 100 μm nozzle before traversing two 2 mm skimmers.

Ions and electrons were extracted in opposite directions by a DC field of 88.7 V/cm, detected by velocity map imaging detectors. and correlated in time so that the photoelectron images could be mass-tagged. At each photon energy, the mass-selected photoelectron spectra were extracted from the images with an Abel inversion algorithm[52]. In an energy scan, the coincident signal as a function of electron kinetic energy and photon energy was recorded and could be reduced to a threshold photoelectron spectrum (TPES) using a previously reported methodology[51]. The photon and electron bandwidths chosen in this work led to a total energy resolution of 20 meV for the TPES presented here. Alternatively, photoelectron spectra (PES) at fixed photon energy could be used as molecular fingerprints when time or stabilities were an issue, at the detriment of energy resolution[53,54].

The photons were provided by the undulator-based DESIRS beamline[55] at the French synchrotron facility SOLEIL. The beamline was set to provide an energy resolution of 8 meV at 9 eV with a flux of $4 \times 10^{12}$ ph/sec. A gas filter filled with Kr cut off the high harmonic radiation from the undulator ensuring spectral purity[56]. The energy scans were corrected by the photon flux as measured by a dedicated photodiode (AXUV, IRD). The energy scale was calibrated using the $4p^5 5s$ $(3/2) \leftarrow 4p^6$ Kr line[57] with a precision of $\pm 4$ meV.

### Chemical synthesis of $C_6H_6$ isomers

**2-ethynyl-1,3-butadiene.** 2E13BD (**VI**), $HC \equiv C-C(=CH_2)-CH = CH$: 2-ethynyl-1,3-butadiene (3-Methylene-1-pentene-4-yne) was synthesized as previously reported[58] by dehydration of 3-methyl-1-penten-4-yn-3-ol (purchased from Aldrich-Merck company and used without further purification) over molecular sieves. Purification of 2E13BD was carried out by trap-to-trap distillation.

**1,3-hexadiene-5-yne.** 13HD5Y (**VIII & XII**), $CH_2 = CHCH = CHC \equiv CH$: 1,3-hexadien-5-yne was synthesized by reaction of 1,5-hexadiyne with a saturated solution of potassium t-butoxide in t-butyl alcohol as previously reported[59]. The crude product in pentane was purified by trap-to-trap distillation.

**Mixture of 1,2-hexadiene-5-yne.** 12HD5Y (**V**), $CH_2 = C = CHC \equiv CH$ and **1,2,4,5-hexatetraene** (1245HT (**II**), $CH_2 = C = CHCH = C = CH_2$). The synthesis of a mixture of 1,2-hexadien-5-yne and 1,2,4,5-hexatetraene was performed in a 3:2 ratio by dimerization of the propargyl radical from propargyl bromide in the presence of cuprous chloride[60,61]. For each synthesis, purity was determined by NMR spectroscopy.

### Computational

To calculate the equilibrium geometries of the different species involved in this study, the exothermicities of the different reactions and the involved transition states, DFT calculations were performed using the M06-2X functional and AVTZ basis set with Gaussian16[62]. The Franck-Condon (FC) factors for the photoionization were calculated using the harmonic approximation for harmonic frequencies and normal modes in the neutral and cationic ground states and the Condon approximation for the dipole moment. The Duschinsky effect was considered using recursive formulae already implemented in the Gaussian16 software package.

## Data availability

All the data presented in this manuscript (threshold photoelectron spectra – experimental and theoretical – as well as mass spectra) are available for download on Zenodo: https://doi.org/10.5281/zenodo.11454182. The electronic supplementary information contains detailed information about the following points: 1) A discussion of the contribution of doubly fluorinated compounds to the *m/z* 78 TPES, including figures for the TPES spectra of these species (TPES for $C_3H_4F$ isomers *m/z* 59 and for $C_3H_4F_2$ isomers *m/z* 77, and adiabatic ionization energies obtained in this work for trans-$CH_2CHCHF$, cis-$CH_2CHCHF$, and $CH_2CFCH_2$), data for the 2-ethynyl-1,3-butadiene and 1,2,4,5-hexatetraene PES spectra; 2) histograms of the weighting factors and their statistical distributions from the Monte-Carlo analysis for reproducing the *m/z* 78 TPES with the reference spectra of the various isomers; 3) Comparison of our experimental TPES with the weighted sum of all contributing reference spectra with the branching ratios determined in the present work (see Table 1 of the main text), as well as with the branching ratios from Miller and Klippenstein study.

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

## Acknowledgements

This work was performed on the DESIRS Beamline at SOLEIL synchrotron under Proposal numbers 99210165, 99210024, 99200162, 99150130 and 20171071. We are grateful to the whole staff of SOLEIL for running the facility. We acknowledge financial support from the French Agence Nationale de la Recherche (ANR) under grant ANR-12-BS08-0020-02 (project SYNCHROKIN). J.-C. G. thanks for financial support the Programme National "Physique et Chimie du Milieu Interstellaire" (PCMI) of CNRS/INSU with INC/INP co-funded by CEA and CNES.

## Author contributions

Helgi Rafn Hrodmarsson: Data Curation, Formal analysis, Investigation, Visualization, Writing – original draft, Writing, review & editing, Gustavo A. Garcia: Data Curation, Investigation, Writing, Validation, review & editing, Lyna Bourehil: Data Curation, Validation, review & editing, Laurent Nahon: Investigation, Project Administration, Resources, Supervision, Writing, review & editing, Bérenger Gans: Data Curation, Formal analysis, Visualization, Writing, review & editing, Séverine Boyé-Péronne: Data Curation, Formal analysis, Visualization, Writing, review & editing, Jean-Claude Guillemin: Data Curation, Resources, Validation, Jean-Christophe Loison: Conceptualization, Data Curation, Formal Analysis, Visualization, Validation, Supervision, Writing, review & editing.

## Competing interests

The authors declare no competing interests.
