## [Peer Review File · Communications Chemistry]

Web links to the author's journal account have been redacted from the decision letters as indicated to maintain confidentiality.

Decision letter and referee reports: first round

22nd April 2024

Dear Dr Hrodmarsson,

Your manuscript titled "The isomer distribution of C₆H₆ products from the propargyl gas-phase recombination investigated by threshold-photoelectron spectroscopy" has now been seen by 3 referees. You will see from their comments below that while they find your work of interest, some important points are raised. We are interested in the possibility of publishing your study in *Communications Chemistry*, but would like to consider your response to these concerns in the form of a revised manuscript before we make a final decision on publication.

We therefore invite you to revise and resubmit your manuscript, taking into account the points raised. Please highlight all changes in the manuscript text file.

To improve the quality of methods and statistics reporting in our papers, we are now asking all authors to complete an editorial policy checklist that verifies compliance with all required editorial policies. Please ensure that the checklist is completed and uploaded with your revised article. All points on the policy checklist must be addressed; if needed, please revise your manuscript in response to these points. Please note that this form is a dynamic 'smart pdf' and must therefore be downloaded and completed in Adobe Reader. Clicking this link will download a zip file containing the pdf.

Editorial policy checklist: <https://www.nature.com/documents/nr-editorial-policy-checklist.pdf> (Download the link to your computer as a PDF.)

In addition, please ensure that the following requirements are met, and that any other relevant checklists are completed and uploaded under the 'Related Manuscript file' type with the revised article.

In the event that your manuscript is accepted, we will provide detailed guidance on our journal policies and formatting. You may however wish to ensure that the manuscript broadly complies with our house style at this stage. See our style and formatting guide (<https://www.nature.com/documents/commsj-phys-style-formatting-guide-accept.pdf>) and checklist (<https://www.nature.com/documents/commsj-phys-style-formatting-checklist-article.pdf>) for reference.

Data availability statements and data citations policy: All *Communications Chemistry* manuscripts must include a section titled "Data Availability" at the end of the Methods section or main text (if no Methods). More information on this policy, and a list of examples, is available at <http://www.nature.com/authors/policies/data/data-availability-statements-data-citations.pdf>.

- Accession codes for deposited data
- Other unique identifiers (such as DOIs and hyperlinks for any other datasets)
- At a minimum, a statement confirming that all relevant data are available from the authors
- If applicable, a statement regarding data available with restrictions
- If a dataset has a Digital Object Identifier (DOI) as its unique identifier, we strongly encourage including this in the Reference list and citing the dataset in the Data Availability Statement.

DATA SOURCES: We strongly encourage authors to deposit all new data associated with the paper in a persistent repository where they can be freely and enduringly accessed. We recommend submitting the data to discipline-specific, community-recognized repositories, where possible and a list of recommended repositories is provided at <http://www.nature.com/sdata/policies/repositories>.

If a community resource is unavailable, data can be submitted to generalist repositories such as [figshare](https://www.figshare.com/) or [Dryad Digital Repository](https://www.dryad.org/). Please provide a unique identifier for the data (for example

Decision letter and referee reports: first round

a DOI or a permanent URL) in the data availability statement, if possible. If the repository does not provide identifiers, we encourage authors to supply the search terms that will return the data. For data that have been obtained from publically available sources, please provide a URL and the specific data product name in the data availability statement. Data with a DOI should be further cited in the methods reference section.

Please use the following link to submit your revised manuscript, point-by-point response to the referees' comments (which should be in a separate document to any cover letter and should ideally refer to any anonymous reviewers using gender neutral they/them/their pronouns) and any completed checklist:

[REDACTED]

We hope to receive your revised paper within three months; please let us know if you aren't able to submit it within this time so that we can discuss how best to proceed. If we don't hear from you, and the revision process takes significantly longer, we will close your file. In this event, we will still be happy to reconsider your paper at a later date, as long as nothing similar has been accepted for publication at Communications Chemistry or published elsewhere in the meantime.

Please do not hesitate to contact me if you have any questions or would like to discuss these revisions further. We look forward to seeing the revised manuscript and thank you for the opportunity to review your work.

Best regards,

Katrin Erath-Dulitz, DPhil
Editorial Board Member
Communications Chemistry
orcid.org/0000-0003-0489-6038

Reviewers' comments:

Reviewer #1 (Remarks to the Author):

The propargyl self-reaction has important significance in chemical reaction kinetics. It may help in understanding the formation of PAHs in the interstellar medium (ISM). The authors of the reviewed manuscript determined the branching ratios of propargyl self-reaction by combining i2PEPICO measurements of C₆H₆ isomers, quantum chemistry calculations, and Franck-Condon factor simulations. The branching ratios determined experimentally have been used to benchmark theoretical calculations.

Both the experimental measurements and computational results are reliable and of high quality. The objective, to accurately determine reaction branching ratios, has been achieved convincingly. The combined experimental/computational methodology developed here will have wide applications in chemical kinetic studies. Furthermore, the manuscript provides valuable insights into the complex chemistry of C₆H₆ isomer formation from the propargyl self-reaction. Therefore, I would recommend the manuscript for publication.

It would be helpful to provide more details on the Monte-Carlo simulation in the supporting information.

The manuscript itself needs proofreading and polishing. For example:

Decision letter and referee reports: first round

- I believe it would be better to use "resonance-stabilization" than "resonant-stabilisation";
- In a couple of places, the authors used "... allots it ...". Is it better to just say "... makes it ..."?
- "...understanding the fundamental reactions that lead to the formation of PAHs and eventually soot particles are of paramount interest..." "are""is".
- "Experimental validation of the above-mentioned model and quantification of the benzene exit channel are challenging..." "are""is".
- "We obtain a good agreement..."  "...obtained...".
- "... ratio determined by their NMR signature."  " a ratio ..."
- "...a remarkable good agreement..."  "... remarkably good ..."

Reviewer #2 (Remarks to the Author):

This manuscript reports a careful study of the 78 u products observed in a low pressure room temperature flow containing propyne and F atoms in helium. The conditions are chosen to favor formation of the propargyl radicals for the study of the self-reaction, an important process leading to the formation of the first aromatic ring in combustion and astrochemical environments. The work is performed using threshold photoelectron spectroscopy at the SOLEIL synchrotron using a double imaging VMI system, and the spectra are compared to their own for isolated candidate isomers or to literature values where available, and to theory. This is certainly state-of-the art and adds significantly to the knowledge base on this important system. I have only a few minor suggestions for the authors after which I recommend publication.

I suggest combining Fig. 4 a and d into a single figure with two colors to represent the two isomers. It is the only case where two different species are compared to a single spectrum and is a bit confusing the way it is shown.

It would be useful to discuss the temperature dependence of the branching further. When they say there is agreement with Klippenstein and Miller, is this at room temperature? Implications for low temperature would also be of interest for astrochemistry.

In a number of cases the English is awkward: particularly in the opening sentences, but also in a few other places (e.g. "criterium").

The references almost all list only the first author. I suspect this is some default in the bibliography software and not a conscious choice and I hope it is not in keeping with the journal standard.

The author contribution statement is from the template and is thus not a contribution statement at all.

Reviewer #3 (Remarks to the Author):

The authors provide an interesting demonstrative application of the use of threshold photoelectron spectroscopy (TPES) to determine the isomer-resolved product distribution for one of the most important reactions in the chemistry of hydrocarbon ring formation chemistry. The work is reasonably well described, informative, and novel. The TPES method holds promise for many further studies of product state distributions. Thus, I recommend it be accepted for publication once the authors respond to a few minor questions I have.

It is not clear to me why the authors use the calculated spectra for their fitting functions rather than the experimentally determined reference spectra. Some detailed explanation of the rationale for that choice would be helpful.

Decision letter and referee reports: first round

I am curious to see what the spectra would have looked if they had simply employed the product distributions predicted by Miller and Klippenstein. A plot of that (perhaps in the supplementary material) could provide a very nice visual illustration of how significant the deviations are.

On line 286, the authors indicate that similar signals imply a fulvene/benzene ratio of 0.5. But on lines 282/283, they indicate that the conversion from signal to ratio instead increases the estimate by a factor of two. I am confused as to why in one case, the conversion increases the ratio, and in the other it decreases the ratio. Furthermore, in Table I, the ratio is 14/11. So, in the end, I am very confused as to what they are saying about the fulvene to benzene ratio.

My interpretation of the purpose of the Monte Carlo simulations was to obtain error bars on the estimated branching ratios. If I understand correctly, the method employed effectively considers single variations in each ratio, but not simultaneous variations in the whole set of branching ratios. If this understanding is correct, then their approach effectively explores the local minimum that they settled on by hand. Is this understanding correct? If it is, they might care to comment on whether or not there might be other local minima for the fits and how one might explore such possibilities.

There are some odd word choices in places – for example, “emblematises”, “agglutination”, “feryl”. Also, the phrase “equilibrium reaction present between 2E13BD and fulvene” makes me think that somehow 2E13BD reacts with fulvene. Presumably, what is meant is that 2E13BD and fulvene equilibrate.

Author Responses: First round

Reviewer #1 (Remarks to the Author):

The propargyl self-reaction has important significance in chemical reaction kinetics. It may help in understanding the formation of PAHs in the interstellar medium (ISM). The authors of the reviewed manuscript determined the branching ratios of propargyl self-reaction by combining i^2 PEPICO measurements of C_6H_6 isomers, quantum chemistry calculations, and Franck-Condon factor simulations. The branching ratios determined experimentally have been used to benchmark theoretical calculations.

Both the experimental measurements and computational results are reliable and of high quality. The objective, to accurately determine reaction branching ratios, has been achieved convincingly. The combined experimental/computational methodology developed here will have wide applications in chemical kinetic studies. Furthermore, the manuscript provides valuable insights into the complex chemistry of C_6H_6 isomer formation from the propargyl self-reaction. Therefore, I would recommend the manuscript for publication.

It would be helpful to provide more details on the Monte-Carlo simulation in the supporting information.

Author response: We have slightly modified the description of the Monte-Carlo procedure in the text, and we have added a paragraph in section 2) of the SI to give further details.

The manuscript itself needs proofreading and polishing. For example:

- I believe it would be better to use “resonance-stabilization” than “resonant-stabilisation”;

Author response: We have made the appropriate change suggested by the reviewer.

- In a couple of places, the authors used “... allots it ...”. Is it better to just say “... makes it ...”?

Author response: In the abstract and we have replaced the word ‘allots’ with ‘grants’. This way the meaning is clearer.

- “...understanding the fundamental reactions that lead to the formation of PAHs and eventually soot particles are of paramount interest...” “are””is”.

Author response: We have made the appropriate change.

- “Experimental validation of the above-mentioned model and quantification of the benzene exit channel are challenging...” “are””is”.

Author response: We have made the appropriate change.

- “We obtain a good agreement...”  “...obtained...”.

Author response: We have made the appropriate change.

- “ ... ratio determined by their NMR signature.”  “ a ratio ...”

Author response: We have made the appropriate change.

- "...a remarkable good agreement..."  "... remarkably good ..."

Author response: We have made the appropriate change.

Reviewer #2 (Remarks to the Author):

This manuscript reports a careful study of the 78 u products observed in a low-pressure room temperature flow containing propyne and F atoms in helium. The conditions are chosen to favor formation of the propargyl radicals for the study of the self-reaction, an important process leading to the formation of the first aromatic ring in combustion and astrochemical environments. The work is performed using threshold photoelectron spectroscopy at the SOLEIL synchrotron using a double imaging VMI system, and the spectra are compared to their own for isolated candidate isomers or to literature values where available, and to theory. This is certainly state-of-the art and adds significantly to the knowledge base on this important system. I have only a few minor suggestions for the authors after which I recommend publication.

I suggest combining Fig. 4 a and d into a single figure with two colors to represent the two isomers. It is the only case where two different species are compared to a single spectrum and is a bit confusing the way it is shown.

Author response: After various trials, we find that combining the two into a single figure fails in clarifying the situation but rather makes things more complicated. We prefer to leave these figures as they are, but clarifications on reference spectra have been added in the "Reference TPES" section of the article to clear up any confusion.

It would be useful to discuss the temperature dependence of the branching further. When they say there is agreement with Klippenstein and Miller, is this at room temperature? Implications for low temperature would also be of interest for astrochemistry.

Author response: We specified that the agreement concerned the temperature of our study (room temperature).

The implications for astrochemistry (very low temperature and very low pressure) are difficult to predict without specific calculations. Indeed, Miller and Klippenstein's study only covers branching ratios for temperatures above 300K and pressures above 1 torr. Moreover, radiative association, not considered in Miller and Klippenstein study, can play an important role at very low temperatures. For Titan, Vuitton *et al.* 2019 (S. Klippenstein is one of the authors) use a high rate for radiative association equal to $6.5e-11 \text{ molecule}^{-1} \cdot \text{cm}^3 \cdot \text{s}^{-1}$ for Titan temperature conditions ($T = 90-150\text{K}$) (reaction 169 in Table B.15 of Vuitton 2019). The study of the $\text{C}_3\text{H}_3 + \text{C}_3\text{H}_3$ reaction at low pressure and low temperature of the dense molecular cloud remains to be done, in particular for the competition between the radiative association and the bimolecular $\text{H} + \text{C}_6\text{H}_5$ pathway.

In a number of cases the English is awkward: particularly in the opening sentences, but also in a few other places (e.g. "criterium").

Author response: We have replaced 'criterium' with 'criterion'.

The references almost all list only the first author. I suspect this is some default in the bibliography software and not a conscious choice and I hope it is not in keeping with the journal standard.

Author response: This is the standard bibliographic format for *Communications in Chemistry* where the author list is abridged if there are more than five authors.

The author contribution statement is from the template and is thus not a contribution statement at all.

Author response: We have added an author contribution statement.

Reviewer #3 (Remarks to the Author):

The authors provide an interesting demonstrative application of the use of threshold photoelectron spectroscopy (TPES) to determine the isomer-resolved product distribution for one of the most important reactions in the chemistry of hydrocarbon ring formation chemistry. The work is reasonably well described, informative, and novel. The TPES method holds promise for many further studies of product state distributions. Thus, I recommend it be accepted for publication once the authors respond to a few minor questions I have.

It is not clear to me why the authors use the calculated spectra for their fitting functions rather than the experimentally determined reference spectra. Some detailed explanation of the rationale for that choice would be helpful.

Author response: We use the calculated spectra in cases where the reference spectra are recorded under conditions too different from our experimental spectrum of $m/z = 78$ from the $C_3H_3 + C_3H_3$ reaction. This is the case for 1245HT (the calculated spectra being in very good agreement with our experimental spectra which is not easy to use as it is mixed with 12HD5Y), and for 2E13BD (our experimental spectrum is recorded with lower resolution and in a different mode (fixed-energy PES, not TPES)). In all other cases (34DMCB, fulvene, 2E13BD, benzene and 13HD5Y), we use experimental spectra. We have clarified this further in the “**Reference TPES**” section of the article.

I am curious to see what the spectra would have looked if they had simply employed the product distributions predicted by Miller and Klippenstein. A plot of that (perhaps in the supplementary material) could provide a very nice visual illustration of how significant the deviations are.

Author response: One figure (Figure S7) has been added in the SI with the simulated spectrum using the branching ratios predicted by Miller and Klippenstein. It clearly shows some deviations from the experimental spectrum above 8.7 eV.

On line 286, the authors indicate that similar signals imply a fulvene/benzene ratio of 0.5. But on lines 282/283, they indicate that the conversion from signal to ratio instead increases the estimate by a factor of two. I am confused as to why in one case, the conversion increases the ratio, and in the other it decreases the ratio. Furthermore, in Table I, the ratio is 14/11. So, in the end, I am very confused as to what they are saying about the fulvene to benzene ratio.

Author response: The referee is entirely correct, and we have rectified the confusion. Note that it does not change the branching ratio values given in Table I.

My interpretation of the purpose of the Monte Carlo simulations was to obtain error bars on the estimated branching ratios. If I understand correctly, the method employed effectively considers single variations in each ratio, but not simultaneous variations in the whole set of branching ratios. If this understanding is correct, then their approach effectively explores the local minimum that they settled on by hand. Is this understanding correct? If it is, they might care to comment on whether or not there might be other local minima for the fits and how one might explore such possibilities.

Author response: In fact, in our method we vary all the ratios simultaneously. The initial (manually adjusted) distribution serves as a starting point. As we vary each distribution by +/- 0.2 (imposing a positive or zero value for each ratio) and as the ratios are lower (or very close to 0.2), we explore virtually all possibilities, and not just a possible local minimum located close to the initial manually-adjusted distribution. To clarify this point, we have slightly modified the description of the Monte-Carlo procedure in the text and added a paragraph in section 2) of the SI.

There are some odd word choices in places – for example, “emblematises”, “agglutination”, “fenyl”. Also, the phrase “equilibrium reaction present between 2E13BD and fulvene” makes me think that somehow 2E13BD reacts with fulvene. Presumably, what is meant is that 2E13BD and fulvene equilibrate.

Author response: We have replaced the word ‘emblematises’ with ‘represents’, the word ‘agglutination’ with ‘agglomeration’, and modified the text in line with the comments made by the referee.

Decision letter and referee reports: second round

22 May 2024

Dear Dr Hrodmarsson,

Your manuscript titled "The isomer distribution of C₆H₆ products from the propargyl gas-phase recombination investigated by threshold-photoelectron spectroscopy" has now been seen again by two of our referees, whose comments appear below. In light of their advice I am delighted to say that we are happy, in principle, to publish a suitably revised version in Communications Chemistry under the open access CC BY license (Creative Commons Attribution v4.0 International License).

We therefore ask that you edit your manuscript to comply with our journal policies and formatting style in order to maximise the accessibility and therefore the impact of your work.

EDITORIAL REQUESTS

* Your manuscript should comply with our policies and format requirements, detailed in our style and formatting guide (<https://www.nature.com/documents/commsj-phys-style-formatting-guide-accept.pdf>).

* Please edit your manuscript according to the editorial requests in the attached table, and outline revisions made in the right hand column. If you have any questions or concerns about any of our requests, please do not hesitate to contact me. It is important that each request be addressed in order to avoid delays in accepting your manuscript. Please upload the completed table with your manuscript files as a Related Manuscript file.

* An updated editorial policy checklist that verifies compliance with all required editorial policies must be completed and uploaded with the revised manuscript. All points on the policy checklist must be addressed; if needed, please revise your manuscript in response to these points. Please note that this form is a dynamic 'smart pdf' and must therefore be downloaded and completed in Adobe Reader. Clicking this link will download a zip file containing the pdf.

Editorial policy checklist: <https://www.nature.com/documents/nr-editorial-policy-checklist.pdf>
(Download the link to your computer as a PDF.)

OPEN ACCESS

Communications Chemistry is a fully open access journal. Articles are made freely accessible on publication under a CC BY license (Creative Commons Attribution 4.0 International License). This license allows maximum dissemination and re-use of open access materials and is preferred by many research funding bodies.

For further information about article processing charges, open access funding, and advice and support from Nature Research, please visit <https://www.nature.com/commschem/about/open-access>

RESUBMISSION

Please refer to our checklist for a full list of the files that must be provided upon resubmission: <https://www.nature.com/documents/commsj-file-checklist.pdf>

Please use the following link to submit your revised files:

[REDACTED]

Decision letter and referee reports: second round

We hope to hear from you within two weeks; please let us know if the process may take longer.

Best regards,

Teresa

Dr Teresa Schauperl

Senior Editor
Communications Chemistry
Heidelberger Platz 3
14197 Berlin
www.nature.com/commschem

orcid.org/0000-0002-0316-1411

on behalf of

Katrin Erath-Dulitz, DPhil
Editorial Board Member
Communications Chemistry
orcid.org/0000-0003-0489-6038

REVIEWERS' COMMENTS:

Reviewer #2 (Remarks to the Author):

I am satisfied with the changes and recommend publication.

Reviewer #3 (Remarks to the Author):

The authors have provided very satisfactory responses to my concerns. I now consider this appropriate for publication in Nature Communications Chemistry.